# PSMA-PET/CT-Based Stereotactic Body Radiotherapy (SBRT) in the Treatment of Uncomplicated Non-Spinal Bone Oligometastases from Prostate Cancer

**DOI:** 10.3390/cancers15102800

**Published:** 2023-05-17

**Authors:** Edoardo Pastorello, Luca Nicosia, Francesco Cuccia, Laura Olivari, Matilde Fiorini, Niccolò Giaj Levra, Rosario Mazzola, Francesco Ricchetti, Michele Rigo, Paolo Ravelli, Salvatore D’Alessandro, Matteo Salgarello, Ruggero Ruggieri, Filippo Alongi

**Affiliations:** 1Advanced Radiation Oncology Department, IRCCS Sacro Cuore Don Calabria Hospital, Cancer Care Center, 37024 Negrar, Italy; 2Francesco Cuccia, Radiotherapy Unit, ARNAS Civico Hospital, 90127 Palermo, Italy; 3Nuclear Medicine Unit, IRCCS Ospedale Sacro Cuore Don Calabria, 37024 Negrar, Italymatteo.salgarello@sacrocuore.it (M.S.); 4Radiation Oncology School, University of Brescia, 25121 Brescia, Italy

**Keywords:** non-spinal bone metastases, SBRT, stereotactic body radiotherapy, SABR, PSMA PET/CT, oligometastases, prostate cancer

## Abstract

**Simple Summary:**

Stereotactic body radiotherapy has a consolidated role in the treatment of oligometastases from prostate cancer. Limited clinical data explored its use in non-spinal bone metastases, as well as details regarding its dose and target definition. We analyzed the outcome of 95 patients treated for 150 non-spinal bone metastases from prostate cancer, aiming to evaluate the local control, the pattern of relapse, and toxicity. The target was represented by the macroscopic disease defined by ^68^Ga-PSMA-PET/CT and CT. The results demonstrated a high local control level and only eight cases of relapse within the same bone. This study provided further evidence supporting the use of SBRT in non-spinal bone metastases from prostate cancer.

**Abstract:**

Background and purpose: Stereotactic body radiotherapy (SBRT) has a consolidated role in the treatment of bone oligometastases from prostate cancer (PCa). While the evidence for spinal oligometastases SBRT was robust, its role in non-spinal-bone metastases (NSBM) is not standardized. In fact, there was no clear consensus about dose and target definition in this setting. The aim of our study was to evaluate efficacy, toxicity, and the pattern of relapse in SBRT delivered to NSBM from PCa. Materials and methods: From 2016 to 2021, we treated a series of oligo-NSBM from PCa with ^68^Ga-PSMA PET/CT-guided SBRT. The primary endpoint was local progression-free survival (LPFS). The secondary endpoints were toxicity, the pattern of intraosseous relapse, distant progression-free survival (DPFS), polimetastases-free survival (PMFS), and overall survival (OS). Results: a total of 150 NSBM in 95 patients were treated with 30–35 Gy in five fractions. With a median follow-up of 26 months, 1- and 3 years LPFS was 96.3% and 89%, respectively. A biologically effective dose (BED) ≥ 198 Gy was correlated with improved LPFS (*p* = 0.007). Intraosseous relapse occurred in eight (5.3%) cases. Oligorecurrent disease was associated with a better PMFS compared to de novo oligometastatic disease (*p* = 0.001) and oligoprogressive patients (*p* = 0.007). No grade ≥ 3 toxicity occurred. Conclusion: SBRT is a safe and effective tool for NSBM from PCa in the oligometastatic setting. Intraosseous relapse was a relatively rare event. Predictive factors of the improved outcomes were defined.

## 1. Introduction

Bones are frequent sites of metastases in advanced prostate cancer (PCa), and stereotactic body radiotherapy (SBRT) has a consolidated role showing high local control levels and limited toxicity [1,2,3,4]. The majority of the evidence relates to spinal bone metastases where radiation dose and volume definition are well defined [5]. In recent years, non-spinal bone metastases (NSBM) are also acquiring clinical interest, even if there is a lack of clinical evidence and no clear consensus on the target volume definition [6]. From a clinical perspective, there are only little data regarding SBRT use in oligometastatic uncomplicated NSBM in contrast to the palliative NSBM setting [7].

In the ablative setting, the correct NSBM lesion identification is a key part of the treatment. In fact, PET/CT and MRI are more accurate than traditional CT imaging and bone scans when identifying macroscopic bone lesions [8]. In PCa in particular, choline PET/CT, and the Prostate Specific Membrane Antigen (PSMA) PET/CT, have improved the detection rate of PCa metastases in all settings (hormone-sensitive and castration-resistant) and at low PSA levels [9,10]. Nevertheless, the identification of microscopical disease remains crucial to reduce peripheral relapse and increasing local disease control. Some guidelines suggest including a clinical target volume (CTV) around NSBM ranging from a few millimeters to more than 1 cm [5]. In the era of precision radiotherapy, technological improvements ensure high precision. Therefore, is it still necessary to use a large treatment volume for bone metastases, or is focal treatment equally safe and effective? In order to evaluate this hypothesis, we analyzed the tolerability and effectiveness of SBRT when delivered to NSBM. The intraosseous pattern of relapse was also evaluated.

## 2. Material and Methods

From July 2016 to November 2021, a series of oligometastatic PCa patients with NSBM treated with SBRT in our Department were retrospectively reviewed. This study was approved by the Institutional Review Board. Patients were included according to the following characteristics: (a) performance status—Eastern Cooperative Oncology Group (PS (ECOG)) ≤ 1; (b) PCa with evidence of NSBM diagnosed by ^68^Ga-PSMA PET/CT and a detectable PSA at the time of the exam; (c) bone lesion ≤ 30 mm; (d) circumferential cortical involvement on CT ≤ 30 mm; (e) Mirels’ score ≤ 7 [11]; (f) maximum local pain ≤ 4/10 measured with a Numeric Pain Rating Scale (NPRS); (g) oligometastatic state including de novo oligo metastatic, oligorecurrent, and oligoprogressive.

^68^Ga-PSMA-11 (HBED-CC)-PET/CT images were performed 60 min after the tracer injection (1.2 MBq/Kg/body weight) from the vertex to the upper portion of the thighs in accordance with European Association of Nuclear Medicine guidelines. PET scans were acquired with a field of view (FOV) diameter of 50 cm and were reconstructed by a Siemens mCT Biopgraph iterative reconstruction algorithm (TrueX PFS + TOF; 21 subsets by 3 interactions; 128 × 128 matrices; 5 mm FWHM Gaussian filter)The lesions were defined according to both biochemical recurrence (PSA rise) and new bone uptake defined on PSMA PET along with morphological CT-based modification (i.e., blastic bone lesions). This imaging information was required for the SBRT prescription. No bone biopsy was performed.

The oligometastatic disease was defined by the ESTRO consensus [12]. Local progression-free survival (LPFS) was defined as the time between the end of SBRT and the radiological diagnosis of in-field relapse. Distant intraosseous relapse (DIR) was defined as the radiological occurrence of new metastases in the same bone segment outside of the treatment field. Distant progression-free survival (DPFS) was defined as the time between SBRT and the radiological diagnosis of distant progression. Polymetastases-free survival (PMFS) was defined as the interval between the end of SBRT and the onset of more than 5 new metastases. The disease-free interval (DFI) was defined as the time between the diagnosis and the occurrence of oligometastatic disease.

Toxicity was recorded according to the Common Terminology Criteria for Adverse Events (CTCAE version 5.0) as acute (within 60 days from SBRT end) and late (more than 60 days after SRT end). Follow-up was performed with PSA every 3 months, and ^68^Ga-PSMA PET-TC was performed in case of a PSA rise after SBRT.

## 3. Treatment Procedure

A simulation of CT 3 mm slice thickness (reconstruction to 1.5 mm thickness) was acquired in the supine position; for the pelvic and lower limbs, patients were immobilized with Combifix and arms on the chest; for the thorax and upper limbs, a thermoplastic mask was used. The gross tumor volume (GTV) was defined on the simulation of CT fused with ^68^Ga-PSMA PET/CT. The planning target volume (PTV) was obtained by adding a 5 mm isotropic margin to the GTV. The plan evaluation ensured at least 95% of the prescribed dose to 95% of the PTV without exceeding more than 107% of the prescribed dose. The treatment was delivered using the Volumetric Modulated Arc Therapy (VMAT) technique with Flattening Filter-Free modality (FFF). Daily positioning was ensured by pre-treatment Cone beam CT. The prescribed dose was 30–35 Gy in 5 daily fractions.

## 4. End-Points and Statistics

The primary endpoint was LPFS. The secondary endpoints were toxicity, the pattern of bone relapse, DPFS, PMFS, and OS. The univariate analysis was performed with the Kaplan–Meier method. The log-rank test was applied to determine the differences between the corresponding curves. The following covariates were evaluated for survival end-points: bone site (pelvic bone, flat bone, long bone), biologically effective dose (BED), concomitant systemic therapy, DFI, and oligometastases number. Univariate and multivariate analyses were performed by the Cox proportional hazards model. We included in the analysis all the clinically relevant variables in the univariate analysis (*p* < 0.2). BED was calculated using an a/b ratio of 1.5 Gy. Statistical analysis was performed using SPSS v20.0 software (IBM software, Armonk, NY, USA). A *p*-value < 0.05 indicated a significant correlation.

## 5. Results

### 5.1. Patients’ Characteristics

From July 2016 to November 2021, 358 metastatic PCa patients were treated with radiotherapy to NSBM at our Institution. Two hundred sixty-three (263) patients were excluded because of the use of palliative doses (i.e., 5 × 4 Gy, 10 × 3 Gy), an absence of ^68^Ga-PSMA PET-CT both at treatment and disease relapse, lesions at risk for bone fracture (i.e., extensive cortical bone interruption), an absence of detectable bone alteration at CT-scans, a follow-up shorter than 6 months, polymetastatic disease, and incomplete clinical history. Therefore, 150 non-spine bone oligometastases in 95 PCa patients represented the study population. SBRT was delivered with a median total dose of 35 Gy (range 30–35 Gy) in five fractions. The median age was 70 years (range 57–87). The median DFI was 17 (range 0–228). The oligometastatic state was: oligorecurrent (77.8%), oligometastatic de novo (13.5%), and oligoprogressive in (8.7%). Patients were treated with 1 oligometastasis (39%), 2 metastases (28.5%), 3 lesions (20%), and 4–5 metastases (12.5%). Non-spine lesions were 86 (57%) pelvic bones, 52 (34.5%) flat bones, and 12 (8.5%) long bones. The patients’ characteristics are summarized in Table 1.

### 5.2. Local Control

The median follow-up was 26 months (range 8–71). In the overall population, the 1-, 2- and 3-year LC rates were 96.3%, 91.8%, and 89% (Table 2). At univariate analysis, the factors correlated with improved LC was BED ≥ 198 Gy1.5 (*p* = 0.007). This factor was confirmed as an independent factor in the multivariate analysis (*p* = 0.031; HR 0.224, 95% CI 0.058–0.875). In particular, 1-, 2- and 3-year LC were 90.8%, 84.1%, and 78.5% for BED < 198 Gy1.5, and 99%, 97.5%, and 93.1% for ≥ 198 Gy1.5. See Table 3. At the last follow-up, nine (6%) lesions locally recurred. Seven of them were in-field, while two were both local and marginal. Eight patients had a distant intraosseous relapse with a 1-, 2-, and 3-year DIR of 97.3%, 94.5%, and 92.5%. In the univariate analysis, no difference in intraosseous relapse occurred by concomitant systemic therapy (*p* = 0.4).

### 5.3. Distant Progression and Polymetastatic Disease

The median DPFS was 12 months, and the 1-, 2- and 3 years of DPFS were rated at 50%, 31% and 20.5%, respectively. In both univariate analysis and multivariate analysis, no factors were related to DPFS. PMFS was 86.2%, 71.7% and 64% at 1-, 2- and 3 years. At univariate analysis, oligorecurrent disease was associated with better PMFS (*p* = 0.003) and maintained a significant correlation at multivariate analysis, both in relation to de novo oligometastatic (*p* = 0.001; HR 0.155; 95%CI 0.052–0.466) and oligoprogressive patients (*p* = 0.007; HR 0.121; 95%CI 0.026–0.566). Concomitant systemic therapy showed a negative trend with PMFS but was not significant at univariate analysis (*p* = 0.07); instead, it had a significant correlation at multivariate analysis (*p* = 0.024; HR 2.969; 95%CI 1.157–7.624). In particular, 1- and 3-year PMFS were 81.6%, 52%, and 91%, 79.7% for patients with or without concurrent systemic hormone therapy at the time of SBRT. BED and DFI also showed a trend with the improvement of DPFS, but neither at univariate analysis nor multivariate analysis did it reach a significant correlation. The OS rate was 98.2% at 3 years. No acute or late ≥ G3 adverse events were recorded during treatment or follow-up visits, including no bone fractures.

## 6. Discussion

SBRT has a consolidated role in the treatment of spinal metastases with a high level of local control, pain relief, and adequate safety profile [13]. The interest in the treatment of NSBM has increased in recent years due to the improvement in systemic treatment and prolonged survival, especially in some diseases, such as prostate, breast, and lung cancer [14]. Importantly, technological advances in radiotherapy (i.e., SBRT) supported by clinical data permits the effective treatment of metastatic foci [15]. Additionally, non-spinal bone metastases are a source of cancer pain and can impair the quality of life and patient autonomy [16]. Therefore, especially in the setting of long-surviving or oligometastatic disease, an ablative approach should be pursued to decrease long-term pain and maintain local control and bone stability [17]. However, the clinical evidence in the treatment of NSBM is limited.

In a recent small retrospective study, 34 patients were treated with SBRT (24–40 Gy in 3–5 fractions) to sacral metastases from different primary tumors. The cumulative incidence of local relapse at 2 years was 16.8% which is higher than the one reported in the present series, which is probably also due to the inclusion of non-prostate metastases that might require higher doses to achieve an ablative effect [18]. The majority of the series are retrospective and reports only resulted in 1 year ranging from 66% and 91.8% [19,20,21,22].

The target definition in spinal metastases was defined more than 10 years ago by Cox et al. in a consensus paper describing the SBRT volume based mainly on surgical criteria [5]. Spinal SBRT comprises the addition of a CTV to eventually control the microscopic disease and reduce the intraosseous relapse. In fact, one of the major fears after focal SBRT was represented by the relapse in the same vertebra, which may limit further radiotherapy due to the overdosage of the spinal cord or the risk of bony fracture [23,24]. In the case of non-spinal BM, the available studies included CTV margins from a few millimeters to 1 cm [6]. On the other side, international guidelines suggested a smaller margin but still the inclusion of a CTV. For example, SEOR guidelines for the treatment of NSBM suggested a 5 mm CTV margin +/−, an extraosseous margin of ≤5 mm in patients with soft tissue cancers, and/or significant disruption of the cortical bone [25]. In another recent guideline, there was no definition of the CTV generation strategy; however, the authors suggested the possibility of giving a low dose to the CTV and a simultaneous boost dose to the GTV [26].

In the present study, we delivered SBRT to the GTV only, which was defined by pretreatment ^68^Ga-PSMA-PET/CT and CT. Clinical results demonstrated a high rate of local control at 2 years that was as high as 91.8% in the overall population and 97.5% in the population treated with BED ≥ 198 Gy1.5. These data are in line with previous evidence reporting the 2-year local control of 94.2% in a series of 38 NSBM treated with SBRT using a CTV margin of 1–2 cm [27]. Moreover, the only factor associated with increased local control in the present study was BED ≥ 198 Gy1.5 (5 × 7 Gy) that seems to represent a good balance between effectiveness and safety in oligometastatic PCa, as also previously documented [28].

Toxicity in our series was limited, with no RT-related bone fractures. In a large retrospective series of NSBM from different primary tumors treated with SBRT, bone fractures occurred in 8.5% of cases with lytic lesions and female gender as predictors of it [19].

To evaluate the robustness of the present tumor treatment strategy (GTV-PTV SBRT), we evaluated marginal and intraosseous relapse. Globally, marginal relapse occurred in two lesions, both treated with a low BED and concomitantly with in-field relapse, while intraosseous relapse occurred in 8 out of 150 lesions. Interestingly, half of those cases occurred more than 1 year after primary SBRT, which could relate both to a secondary metastatic wave or previous inactive microscopic disease rather than an intraosseous spread of the treated metastases. Despite the low rate of marginal relapse, we were not able to assess whether the VMAT dose bath could have contributed to a reduction in the risk of peripheral relapse by acting as a “virtual CTV”. This concept was already explored in the brain stereotactic radiosurgery (SRS) context, where brain regions receiving less than 1 Gy from previous SRS had a higher incidence of developing new brain metastases [29]. Overall, intraosseous relapse was efficaciously treated with salvage SBRT with no toxicity.

In conclusion, one of the major issues regarding NSBM remains the target definition. Chapman et al. evaluated interobserver variability in the contouring of NSBM from PCa with different imaging modalities. They demonstrated a higher consistency when using combined contouring modalities (CT + PET and CT + PET + MR) [30]. Additionally, in the study of Ilamurugu et al. [31], it was reported that MR fusion might increase GTV consistency by significantly reducing the Dice Similarity Index and Geographical Miss Index when compared to CT-based contouring alone. Lastly, a larger study compared the interobserver agreement in the GTV definition between CT, MR, and PET, showing the highest consistency in MR-based and PET-based contouring but also no large difference between MR- and PET-based contouring [6]. In the present study, the target volume was homogeneously defined with a combined modality of ^68^Ga-PSMA-PET/CT and CT, as previously described [32,33,34]. This modality was also commonly applied to lung and head-and-neck tumors where PET contouring criteria were better defined [35]. The clinical results of the present study supported the applicability and reproducibility of this contouring modality.

The limits of the study include, in particular, the retrospective nature of the assessment of late toxicity and the concomitant use of systemic therapy in a subgroup of patients, which might have limited the pattern of relapse analysis. Points of strength include the homogeneity of the case series in terms of the clinical setting, contouring strategy, and treatment schedules.

## 7. Conclusions

This study provides further evidence supporting the use of SBRT in NSBM in prostate cancer. Local control is high and comparable to other oligometastatic sites. The treatment of the sole macroscopic disease is not associated with an unacceptable rate of intraosseous relapse. This latter might be treated with salvage radiotherapy with no unexpected toxicity. Further studies are needed to assess late toxicity and its combination with systemic therapies.

## Figures and Tables

**Table 1 cancers-15-02800-t001:** Patients’ characteristics (*n* = 95).

Median age (range)	70 (54–87)
Oligometastases number	
● 1	37 (39%)
● 2	27 (28.5%)
● 3	19 (20%)
● 4	9 (9.5%)
● 5	3 (3%)
Median disease-free interval (months) (range)	17 (0–228)
Oligometastases type	
● Oligorecurrence	74 (77.8%)
● Oligometastatic de novo	13 (13.5%)
● Oligoprogression	8 (8.7%)
Bone site (*n* = 150)	
● Ileum	50 (33.3%)
● Rib	34 (22.7%)
● Sacrum	20 (13.5%)
● Pubis	12 (8%)
● Femur	9 (6%)
● Scapula	8 (5.5%)
● Sternum	6 (4%)
● Clavicle	4 (2.5%)
● Ischium	4 (2.5%)
● Omerus	3 (2%)
Concurrent hormone therapy	
● Yes	49 (51.5%)
● No	46 (48.5%)
Median SBRT total dose (Gy) (range)	35 (30–35)
Median dose per fraction (Gy) (range)	7 (6–7)
Median follow-up months (range)	26 (8–71)
SBRT: stereotactic body radiotherapy	

**Table 2 cancers-15-02800-t002:** Survivals.

	1 Year	2 Year	3 Year
LPFS	96.3%	91.8%	89%
DPFS	50%	31%	20.5%
PMD	86.2%	71.7%	64%
OS	98.2%	98.2%	98.2%

LPFS: local progression-free survival; DPFS: distant progression-free survival; PMD: polymetastaatic disease; OS: overall survival.

**Table 3 cancers-15-02800-t003:** Uni-and multivariate analysis.

Covariates	Univariate	Multivariate
Bone site (base = pelvic bone)	0.163	Flat bone: 0.945
		Long bone: 0.95
Concomitant systemic therapy	0.318	-
BED ≥ 198 Gy1.5	0.007	0.031 (HR 0.224; 95%CI 0.058–0.875)
	DPFS	
	Univariate	Multivariate
DFI < 48 months	0.45	-
Oligometastases number	0.83	-
Bone site (base = pelvic bone)	0.42	-
Oligometastases type (base = oligorecurrent)	0.65	-
BED ≥ 198 Gy1.5	0.45	-
Concomitant systemic therapy	0.21	-
	PMD	
	Univariate	Multivariate
Oligometastases type (base = oligorecurrent)	0.003	De novo: 0.001 (HR 0.155; 95%CI 0.052–0.466)
		Oligoprogressive: 0.007 (HR 0.121; 95%CI 0.026–0.566)
Bone site (base = pelvic bone)	0.49	-
Concomitant systemic therapy	0.07	0.024 (HR 2.969; 95%CI 1.157–7.624)
Oligometastases number	0.34	-
BED ≥ 198 Gy1.5	0.12	0.24 (HR 0.610; 95%CI 0.264–1.414)
Local control	0.44	-
DFI < 48 months	0.14	0.15 (HR 0.530; 95%CI 0.223–1.258)

LPFS: local progression-free survival; BED: biological effective dose; HR: hazard ratio; DPFS: distant progression-free survival; DFI: disease-free interval; PMD: polymetastatic disease.

## Data Availability

The data can be shared up on request.

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
