# Peer review of "PSMA-PET/CT-Based Stereotactic Body Radiotherapy (SBRT) in the Treatment of Uncomplicated Non-Spinal Bone Oligometastases from Prostate Cancer"

_cancers, 2023, doi:10.3390/cancers15102800_

Round 1

Reviewer 1 Report

This paper "PSMA-PET/CT-based stereotactic body radiotherapy (SBRT) in the treatment of uncomplicated non-spinal bone oligometastases from prostate cancer" investigates the roles of SBRT in therapy of non spinal-bone metastases (NSBM) in patients with bone oligometastic prostate cancer. The aim of the study is clearly defined in the paper.

However, I have following comments: 

- There is no clarity on how bone lesions detected by PSMA-PET/CT were classified as suspicious or metastatic: Was there a cutoff value for PSMA uptake (SUV values)? Was PSMA uptake in the bone lesion compared with uptake in the liver? Was a bone biopsy performed. 

- It was not mentioned with which radioactive substance the PSMA peptide was labeled in these PET/CT examinations? 68Ga or 18F?

It is known that PET examinations with 18F-PSMA are associated with nonspecific bone enhancement, sometimes interpreted as suspicious bone lesions. Please discuss this point.

- The imaging modalities for follow-up of these patients and their lesions are also unclear. Were multiple PSMA PET examinations performed in these patients? Interval between examinations? Please discuss the time to PSMA-PET response after radiotherapy.   

- In general, the paper contains too many abbreviations that make it difficult for the reader to follow the topics. Please reduce them. 

Author Response

Dear Reviewer, thank you for your helpful assistance for improving the quality of our manuscript. You can find attached, a word file including the responses to all your comments. Do not hesitate on contacting us for any further information needed. 

Kind Regards

Reviewer 2 Report

The authors have investigated the role of SBRT in NSBM from PCa. The primary endpoint was LPFS and the secondary endpoint were toxicity, pattern of bone relapse, DPFS, PMFS and OS.  Overall, the paper is well written. Some specific comments are provided below-

1) Abstract, line 1: "consolidate" should be "consolidated".

2) Materials and Methods: Was any IRB or institutional approval was obtained for this study? If so, please add the approval/study number.

3) Treatment Procedure: CT slice thickness 3mm is too thick for SBRT planning. Commonly, 2mm or thinner (1.5mm) is recommended.

4) Was the treatment delivered daily or every other day? SBRT, high dose per Fx, is preferably delivered in every other day.

5) Please be consistent with the Rx dose. Some place 30-35Gy, and one place it is 30-36Gy (in results section; medial 35Gy). In Table 1: median dose is 33Gy, range 18-40Gy. I am not sure what Rx dose was used for how many patients. 

6) Hope all patients are male. No mention about the gender/sex of the subjects; better to mention it.

7) Table 1: Is there any reason to make the 1st row in bold?

8) Result (abstract): Follow-up for 26mo (median), 36mo (max); in Table-1 the FU 24mo (median), 65mo (max); in section 5.2 --- range for FU 8-71mo. Inconsistent numbers. Please check all your data carefully.

9) Is 24/26mo FU long enough to determine the efficacy of the treatment?

10) Discussion: Again, "consolidate" should be "consolidated". 

11) Discussion: "In last years, non-spinal BM ..." -- appears awkward, please rephrase it. 

12) "Importantly, technological advanced in radiotherapy (i.e. SBRT) and a better radio-biology understanding allow ....." -- unfortunately, radiobiology of SBRT is not yet well understood.

Author Response

(The authors gave the same response as above.)

Reviewer 3 Report

The authors may be complimented for their efforts to describe the radiotherapeutic parameters and related outcome in these patients. However it seems that different patient groups (primary PCa, HS BCR and CRPC) are mixed together, while 50% used hormones which may introduce a major bias during the relatively short FU period of median 26 months. No detail has been given about the PSMA PET imaging which was the major drive for therapeutic decisions. Furthermore, the ways the outcome parameters were measured are not described, such as follow-up imaging, PSA levels, etc. The conclusions of the research are therefore hard to appreciate. 

See below for some specific comments.

Abstract:

Background is unclear which is partly related to the incorrect English language style which needs improvement.

The definition of the primary endpoint is not clearly explained and therefore the conclusion is hard to understand.

Introduction:

English language style in introduction is suboptimal. Rest of article language is okay

Why would NSBM be so different than SBM in terms of dosing and volume definition as well as prognosis in PCa? Do we really need other evidence than from SBM? Please explain better (part of discussion should be placed in introduction for better understanding).

Ref 8 is dated research and FDG PET is not applicable to PCa so this is confusing here, better leave this out. 

From refs 9 and 10 and further on it is not clear which setting is described: primary PCa? BCR? CRPC? Performance and evidence of PSMA PET in these groups is different as well as the effectiveness of oligometastatic disease and its treatments.

M&M:

Why did these patients get a PSMA PET, was this per-protocol and if so what were there general indications for PSMA PET in the different settings? Describe.

There is no description of PSMA PET protocol and image interpretation. Describe.

Was NSBM diagnosis bases on (clinical) PSMA PET report, multidisciplinary consensus, or else? Describe.

Bone lesion <=30 mm based on….. axial CT longest diameter in MPR mode? Describe.

How were the outcome parameters (DIR, DPFS etc) measured? It is not in M&M.

50% of patients had concurrent hormonal therapy, for how long is not described but probably largely overlapping the median FU duration of 26 months. So it is hard to tell whether effects of treatment are related to the radiotherapy or hormones. This seems a major bias.  

Author Response

Dear Reviewer, thank you for your helpful assistance for improving the quality of our manuscript. You can find attached, a word file including the responses to all your comments. Do not hesitate on contacting us for any further information needed. 

#3
The authors may be complimented for their efforts to describe the radiotherapeutic parameters and related outcome in these patients. However it seems that different patient groups (primary PCa, HS BCR and CRPC) are mixed together, while 50% used hormones which may introduce a major bias during the relatively short FU period of median 26 months. No detail has been given about the PSMA PET imaging which was the major drive for therapeutic decisions. Furthermore, the ways the outcome parameters were measured are not described, such as follow-up imaging, PSA levels, etc. The conclusions of the research are therefore hard to appreciate. 

Authors’ reply: thank-you for your comment

See below for some specific comments.
Abstract:
Background is unclear which is partly related to the incorrect English language style which needs improvement.
The definition of the primary endpoint is not clearly explained and therefore the conclusion is hard to understand.

Authors’ reply: the text of the abstract was edited according to your comment, as follows: “Stereotactic body radiotherapy (SBRT) has a consolidated role in the treatment of bone oligometastases from prostate cancer (PCa). While the evidence for spinal oligometastases SBRT is robust, its role in non spinal-bone metastases (NSBM) is not standardized. In fact, there is no clear consensus about dose and target definition in this setting. The aim of our study was to evaluate efficacy, toxicity, and pattern of relapse of SBRT delivered to NSBM from PCa.”. Moreover, we defined the endpoints in the abstract as follows “The primary endpoint was local progression-free survival (LPFS). The secondary endpoints were toxicity, pattern of intraosseous relapse, distant progression-free survival (DPFS), polimetastases-free survival (PMFS), and overall survival (OS).”

Introduction:
English language style in introduction is suboptimal. Rest of article language is okay

Authors’ reply: We edited the introduction in a more fluent form. We hope it is ok now

Why would NSBM be so different than SBM in terms of dosing and volume definition as well as prognosis in PCa? Do we really need other evidence than from SBM? Please explain better (part of discussion should be placed in introduction for better understanding).

Authors’ reply: Contouring of SBM is well established and Cox guidelines are the current clinical practice, instead there is no clear consensus on target definition for NSBM. In fact, it is not clear whether a CTV is needed and how large this should be to include eventually the microscopic disease. It is clear that Cox guidelines cannot be directly applied from spine to non-spine bone metastases since they are based on spinal surgical margins (see: Int J Radiat Oncol Biol Phys. 2012 Aug 1;83(5):e597-605. doi: 10.1016/j.ijrobp.2012.03.009.). Furthermore, SBRT dose might be different from SBM because of the absence of the spinal cord in NSBM as dose-limiting organ at risk. Nevertheless, we explained this concept in the introduction. We hope that this current version might be fine.

Ref 8 is dated research and FDG PET is not applicable to PCa so this is confusing here, better leave this out. 

Authors’ reply: thank-you for your correction. We selected another ref. that should fit more: Zhou J, Gou Z, Wu R, Yuan Y, Yu G, Zhao Y. Comparison of PSMA-PET/CT, choline-PET/CT, NaF-PET/CT, MRI, and bone scintigraphy in the diagnosis of bone metastases in patients with prostate cancer: a systematic review and meta-analysis. Skeletal Radiol. 2019 Dec;48(12):1915-1924. doi: 10.1007/s00256-019-03230-z. Epub 2019 May 24. PMID: 31127357.

From refs 9 and 10 and further on it is not clear which setting is described: primary PCa? BCR? CRPC? Performance and evidence of PSMA PET in these groups is different as well as the effectiveness of oligometastatic disease and its treatments.

Authors’ reply: Ref 9 refers to castration-sensitive prostate cancer patients, instead ref 10 refers to castration-resistant oligometastatic patients, so these two references should be related to all patients included in our study. We explained it better in the text, hoping it is clearer now.

M&M:
Why did these patients get a PSMA PET, was this per-protocol and if so what were there general indications for PSMA PET in the different settings? Describe.

Author’s reply: thank-you for your careful revision that gives us the occasion to explain this debated argument. PSMA PET was performed according to usual clinical practice in our department and to the EAU Guidelines that support the use of PSMA in case of detectable level >0.2 ng/ml (see https://uroweb.org/guidelines/prostate-cancer/chapter/treatment). We edited inclusion criteria as follows: “PCa with evidence of NSBM diagnosed by 68Ga-PSMA PET/CT and a detectable PSA at the time of the exam”. In particular, in the de novo oligometastatic the PSMA was the staging exam, while in the oligorecurrence setting the PSA threshold of 0.2 ng/ml was required for PSMA7PET indication to define biochemical recurrence, as indicated in EAU guidelines, while in the oligoprogressive setting a PSA raise was considered. Anyway, the end-point of the study was local control, so all the patients were treated to active disease sites.

There is no description of PSMA PET protocol and image interpretation. Describe.

Was NSBM diagnosis bases on (clinical) PSMA PET report, multidisciplinary consensus, or else? Describe.

Author’s reply: In our Hospital PSMA PET-TC is performed using 68Ga. The diagnosis was based first on PSMA PET, but treated lesions were all visible on CT images, that was required for target definition, to avoid false positive from PSMA PET. This issue was also explained in the discussion, evaluating the role of PET and MR in target definition for bone metastases.

Bone lesion <=30 mm based on….. axial CT longest diameter in MPR mode? Describe.

Author’s reply: CT simulation was acquired in 3 mm thickness, but 1.5 mm reconstruction was performed. The GTV was defined on the simulation CT fused with 68Ga-PSMA PET/CT.

Lesions were contoured only on the axial plane and checked for accuracy on sagittal and coronal, as clinical practice in radiotherapy Departments.

How were the outcome parameters (DIR, DPFS etc) measured? It is not in M&M.

Author’s reply: Thank you for your comment. Clinical outcome were defined by PET imaging. Follow-up included PSA measurement and PET PSMA was required in case of PSA raise.

 50% of patients had concurrent hormonal therapy, for how long is not described but probably largely overlapping the median FU duration of 26 months. So it is hard to tell whether effects of treatment are related to the radiotherapy or hormones. This seems a major bias.  

Author’s reply: the primary aim of the study was local control, so the follow-up was powered for this end-point, rather than for long-term survival (i.e. OS). Anyway, recognizing this as a possible confounding factor we already included it into the multivariate analysis, demonstrating no correlation with local control and distant progression, but only in the polymetastatic progression (0.024 (HR 2.969; 95%CI 1.157-7.624)). Moreover, we have already discussed this issue as a possible limitation of the study in the discussion as follows: “Limits of the study are the retrospective nature in particular for the assessment of late toxicity, and the concomitant use of systemic therapy in a subgroup of patients which might have limited the pattern of relapse analysis.”.

Kind Regards

Round 2

Reviewer 1 Report

The authors should incorporate their response to the reviewers into the paper. Please address these 2 points in your manuscript. 

- There is no clarity on how bone lesions detected by PSMA-PET/CT were classified as suspicious or metastatic: Was there a cutoff value for PSMA uptake (SUV values)? Was PSMA uptake in the bone lesion compared with uptake in the liver? Was a bone biopsy performed.

- It is known that PET examinations with 18F-PSMA are associated with nonspecific bone enhancement, sometimes interpreted as suspicious bone lesions. Please discuss this point.

Author Response

On the behalf of all author I want to thank the reviewers for their comments. Please find hereafter point-by-point reply. We believe that the quality of the manuscript have increased and that can be considered for publication in the current form.

The authors should incorporate their response to the reviewers into the paper. Please address these 2 points in your manuscript. 

- There is no clarity on how bone lesions detected by PSMA-PET/CT were classified as suspicious or metastatic: Was there a cutoff value for PSMA uptake (SUV values)? Was PSMA uptake in the bone lesion compared with uptake in the liver? Was a bone biopsy performed.

- It is known that PET examinations with 18F-PSMA are associated with nonspecific bone enhancement, sometimes interpreted as suspicious bone lesions. Please discuss this point.

Authors’ reply: the text was edited according to your suggestions. We hope it’s fine now. “The lesions were defined according to both biochemical recurrence (PSA rise) and new bone uptake defined on PSMA PET along with morphological CT-based modification (i.e. blastic bone lesions). Those imaging information were both required for SBRT prescription. No bone biopsy was performed.”

Reviewer 3 Report

Introduction and purpose of the study is clearer now, as well as the readability of the whole paper.

My main concerns are still in the M&M and results section regarding the non-radiotherapy related part, in order to be able to weigh the scientific soundness of the study. There is too little information regarding

-the selection process of patients (identification, completeness of data, extraction of data from hospital system or else?, database construction),

-the PSMA PET protocol (dosage, incubation time, scanners, reconstruction protocols, scan range, ce or non-ce CT etc etc),

-the interpretation of the initial and follow-up PSMA PET scans and other imaging (who did this, which criteria, how were equivocal lesions dealt with),

-how many patients adhered to the PSA and PSMA PET follow-up and how many lost to follow-up.

Possibly a flow-diagram can be made regarding the selection process and follow-up, making this more clear.

Furthermore, Ga-PSMA is not a tracer, there multiple Ga-PSMA tracers such as Ga-PSMA-11 (HBED-CC), I&T, 617.... Just mention the specific variant in M&M and refer to 'PSMA PET' in the rest of the article, that's fine.

Although the PSMA PET is primarily a tool in this paper and not so much the subject, however I would advise to have a nuclear medicine specialist on board to provide some of above mentioned details.

I look forward to see an updated manuscript.

Author Response

On the behalf of all author I want to thank the reviewers for their comments. Please find hereafter point-by-point reply. We believe that the quality of the manuscript have increased and that can be considered for publication in the current form.

Introduction and purpose of the study is clearer now, as well as the readability of the whole paper.

My main concerns are still in the M&M and results section regarding the non-radiotherapy related part, in order to be able to weigh the scientific soundness of the study. There is too little information regarding

-the selection process of patients (identification, completeness of data, extraction of data from hospital system or else?, database construction),

Authors’reply: The text was edited according to your comments, as follows: “From July 2016 to November 2021, 358 metastatic PCa patients were treated with radiotherapy to NSBM at our Institution. Two hundred sixty-three (263) patients were excluded because: use of palliative doses (i.e. 5 x 4 Gy, 10 x 3 Gy), absence of 68Ga-PSMA PET-CT both at treatment and disease relapse, lesions at risk for bone fracture (i.e. extensive cortical bone interruption), absence of detectable bone alteration at CT-scans, having a follow-up shorter than 6 months, polymetastatic disease,  incomplete clinical history. Therefore, 150 non-spine bone oligometastases in 95 PCa patients represented the study population.”

-the PSMA PET protocol (dosage, incubation time, scanners, reconstruction protocols, scan range, ce or non-ce CT etc etc),

Authors’reply: the text was edited according to a nuclear medicine doctor’s revision, as follows: “68Ga-PSMA-11 (HBED-CC)-PET/CT images were performed 60 minutes after tracer injection (1.2 MBq/Kg/body weight) from the vertex to the upper portion of the thighs in accordance with European Association of Nuclear Medicine guidelines. PET scans were acquired with a field of view (FOV) diameter of 50cm and reconstructed by Siemens mCT Biopgraph interative reconstruction algorithm (TrueX PFS + TOF; 21 subsets by 3 iteractions; 128 x 128 matrix; 5mm FWHM Gaussian filter)”

-the interpretation of the initial and follow-up PSMA PET scans and other imaging (who did this, which criteria, how were equivocal lesions dealt with),

Authors’reply: PET imaging were checked by nuclear medicine doctor experienced in PSMA-PET. The follow-up was already described in MM, as well as further PET indication. PET was performed only in case of PSA raise, since there is no indication to prescribe metabolic imaging in patients with undetectable PSA. So no routine PET was prescribed.

-how many patients adhered to the PSA and PSMA PET follow-up and how many lost to follow-up.

Authors’reply: as described in the above mentioned comment we excluded patients with uncomplete follow-up. Only patients with diagnostic PET were included, as well as those with evaluation PET at the relapse

Possibly a flow-diagram can be made regarding the selection process and follow-up, making this more clear.

Authors’reply: we included the selection criteria in methods section as previously mentioned for readinedd. We hope that this form might be clear as well. Thank-you

Furthermore, Ga-PSMA is not a tracer, there multiple Ga-PSMA tracers such as Ga-PSMA-11 (HBED-CC), I&T, 617.... Just mention the specific variant in M&M and refer to 'PSMA PET' in the rest of the article, that's fine.

Authors’reply: Thank you for your careful clarification. Please refer to the comment above: “68Ga-PSMA-11 (HBED-CC)-PET/CT images were performed 60 minutes after tracer injection (1.2 MBq/Kg/body weight) from the vertex to the upper portion of the thighs in accordance with European Association of Nuclear Medicine guidelines. PET scans were acquired with a field of view (FOV) diameter of 50cm and reconstructed by Siemens mCT Biopgraph interative reconstruction algorithm (TrueX PFS + TOF; 21 subsets by 3 iteractions; 128 x 128 matrix; 5mm FWHM Gaussian filter)”

Although the PSMA PET is primarily a tool in this paper and not so much the subject, however I would advise to have a nuclear medicine specialist on board to provide some of above mentioned details.

I look forward to see an updated manuscript.

Authors’reply: dear reviewer we let a nuclear medicine doctor revise the manuscript and included among the authors. Thank-you for your constructive comment. We hope that the quality of the manuscript have increased and can be accepted in the actual form.

Round 3

Reviewer 3 Report

The manuscript has been updated and the methods / results are okay now and sufficient to support the conclusion.

Please pay some last attention to spelling: "iteration", not "iteraction". "Consolidated", not "consolidate" in Simple Summary.